# Ion Chromatography–High-Resolution Mass Spectrometry Method for the Determination of Bromide Ions in Cereals and Legumes: New Scenario for Global Food Security

**DOI:** 10.3390/foods11162385

**Published:** 2022-08-09

**Authors:** Rachele Rocchi, Roberta Rosato, Mirella Bellocci, Giacomo Migliorati, Rossana Scarpone

**Affiliations:** Istituto Zooprofilattico Sperimentale dell’Abruzzo e del Molise, “G. Caporale”, 64100 Teramo, Italy

**Keywords:** bromide ion, fumigation, cereals and legumes, IC-HRMS, fast analysis

## Abstract

The new scenario for global food production and supply is decidedly complex given the current forecast of an increase in food fragility due to international tensions. In this period, exports from other parts of the world require different routes and treatments to preserve the food quality and integrity. Fumigation is a procedure used for the killing, removal, or rendering infertile of pests, with serious dangers to human health. The most-used fumigants are methyl bromide and ethylene dibromide. It is important to bear in mind that the soil may contain bromide ions naturally or from anthropogenic source (fertilizers and pesticides that contain bromide or previous fumigations). Different methods (titrimetric, spectrophotometric, and fluorometric approaches) are available to rapidly determine the amount of bromide ion on site in the containers, but these are non-specific and with high limits of quantification. The increasing interest in healthy food, without xenobiotic residues, requires the use of more sensitive, specific, and accurate analytical methods. In order to help give an overview of the bromide ion scenario, a new, fast method was developed and validated according to SANTE 11312/2021. It involves the determination of bromide ion in cereals and legumes through ion chromatography–Q-Orbitrap. The extraction was performed by the QuPPe method, but some modifications were applied based on the matrix. The method described here was validated at four different levels. Recoveries were satisfactory and the mean values ranged between 99 and 106%, with a relative standard deviation lower than 3%. The linearity in the matrix was evaluated to be between 0.010 and 2.5 mg kg^−1^, with a coefficient of determination (R^2^) of 0.9962. Finally, the proposed method was applied to different cereals and legumes (rice, wheat, beans, lentils pearled barley, and spelt) and tested with satisfactory results in EUPT-SMR16 organized by EURL.

## 1. Introduction

A new scenario for global food production and supply has appeared due to the implications of current international crises, the COVID-19 pandemic, and climate change, the latter having a direct impact on food systems and food security. Moreover, the war in Ukraine has aggravated existing tensions on the agricultural commodities market. Since the end of 2021, prices for commodities such as grains and vegetable oils have reached unprecedented highs, but now the international tension has made prices soar even higher. This tension changed trade from the major exporting regions and it has been compensated for by other countries [1,2].

In summary, in the last two years, COVID-19 restrictions have affected the activities of many commercial enterprises worldwide, including those involved in the food supply chain, raising many challenges to global food security. Today’s happenings in Russia and Ukraine add another significant challenge, together with climate change, which can affect food availability, access, utilization, and the stability of the different food commodities. Constrictions at any point can lead to food insecurity through the activities of the food system, including food production, transportation, and storage [3,4].

Cereals and legumes are prone to a short shelf life if not given proper treatment [5]. The qualitative and quantitative losses in cereal and legume commodities, starting from harvest to consumption, are huge, and a postharvest treatment/technology is fundamental to preserve losses and increase food security. Fumigation is a very active pest control technique; its practice is a procedure for killing, removing, or rendering infertile pests (FAO, 1990; revised FAO, 1995). The most-used fumigants are carbon disulfide (CS_2_), carbon tetrachloride (CCl_4_), phosphine, dibromoethane, dichloroethane, methyl bromide, and some volatile organophosphorus compounds, such as dichlorvos and pirimiphos-methyl [6]. Among them, methyl bromide is used to control fungi, nematodes, and weeds, both outdoors and indoors; it is also used in space fumigation of food commodities (e.g., grains) and in storage facilities (such as mills, warehouses, vaults, ships, and freight cars) to control insects and rodents. In 2005, the use and sale of methyl bromide as a pesticide/fumigant in the EU was banned apart from quarantine and shipping uses, according to the procedure described in Annex I of the Commission Decision 2005/359/EC [7]. However, it was finally banned with a derogation use until 31 December 2020 [8], for its adverse effects on human health, on the environment, and role in stratospheric ozone depletion [9,10,11].

After the post-harvest fumigation, methyl bromide can be found in the commodities for several days after the end of the treatment. It is transformed into methylated products and bromide ion by reaction with components of treated foodstuffs [12].

In the literature, several works reported the determination of bromide ion in foods. In particular, the proposed methods are the red phenol method [13], ion exchange chromatography (IEC) with a conductivity detector [14], ion chromatography with a conductivity detector [15,16], inductively coupled plasma–mass spectrometry (ICP-MS) [17], gas chromatography with an electron capture detector (GC-ECD) [18], and hydrophilic interaction liquid chromatography and mass spectrometry [19].

The bromide ion, such as other polar ion pesticides (fosetyl-Al, chlorate, and perchlorate), often occur as residues in food, but are not included in multiresidue methods for its non-amenability to reverse-phase chromatographic separation and poor recoveries. For this reason, the European Reference Laboratory (EURL, EU Reference Laboratory) has introduced the Quick Polar Pesticides (QuPPe) Method [20] for single residue methods (EURL-SRM) using acidified methanol and liquid chromatography coupled to tandem mass spectrometry (LC-MS/MS), which has enabled more laboratories to conduct analysis for these polar pesticides in food and biological samples [21,22,23]. This proposed method indicates the use of different stationary phases as graphitized porous carbon (Hypercarb) [24], hydrophilic interaction liquid chromatography (HILIC) [25,26], or ionic exchange. In recent years, other approaches have been reported, such as the use of supercritical fluid chromatography [27], the use of both HILIC and C18 parallel columns [28], and ion chromatography [29,30,31].

Different studies reported on the toxicity of bromide ions, such as mental and neurological disturbances, which are the most common and prominent features [6,32,33]. It is also reported that at enhanced bromide intake, bromine replaces iodine in the rat thyroid and probably remains as bromide ion [34,35]. EFSA considers reviewing the toxicological profile of bromide ions because it is not sufficiently supported by data [36].

Residues in crops were revealed to be not really useful as an indicator of methyl bromide soil fumigation practices. Therefore, the bromide ion may also occur naturally in food and feed and the currently available data are insufficient to determine whether it derives from the pesticide use of methyl bromide or whether it originates from the natural occurrence of bromide ion. The bromide ion naturally occurs in soil, especially in seawater. The concentration of this ion ranged from 1 to 20 mg kg^−1^ [37].

Consequently, it is really relevant to estimate the residues of bromide ion in commodities proposed for consumption to evaluate the consumers’ exposure [38]. It is necessary to collect data to increase the knowledge about the background of bromide ion and for this reason a method to determine the analyte in a routine laboratory quickly would be useful.

This paper aims to develop a new, fast method to determine bromide ion residues in cereals and legumes, using suppressed ion chromatography coupled to high-resolution mass spectrometry (IC-HRMS). The proposed method will help to monitor the compliance with respect to the regulations in force on foods in transit and to evaluate the range of bromide ion content naturally present in cereals and legumes. The European Commission has included in its national monitoring program bromide ion in several commodities in order to collect data [39]. Regulation (EU) 2021/601 [40] requires for the year 2023 the determination of this ion in rice matrices.

## 2. Materials and Methods

### 2.1. Chemicals

The bromide ion reference standard was acquired from Sigma-Aldrich (Saint Louis, MO, USA) with a purity of about ≥99.7% and the internal labelled standard Perchlorate ^18^O_4_ from CIL (Cambridge Isotope Laboratories, Tewksbury, MA, USA). Acetonitrile (MeCN), methanol (MeOH), water (H_2_O), formic acid (HCOOH), and sodium hydroxide (NaOH), all LC-MS grade, were purchased from Sigma Aldrich (Steinheim, Germany). Finally, Pierce LTQ Velos ESI Negative and Positive Ion Calibration Solution were provided by Thermo Fisher Scientific (Waltham, MA, USA).

### 2.2. Sampling

There were 34 samples collected and divided in two groups: cereals and legumes. For cereals, the matrices analyzed were rice (n = 12), wheat (n = 3), barley (n = 1), oats (n = 1), spelt (n = 1), quinoa (n = 1), corn (n = 1), and a mix of cereals (n = 1); for legumes, beans (n = 5), lentils (n = 4), and chickpeas (n = 4). All samples were collected according to the EU regulatory framework of official control activity along the agri-food chain and from border inspection posts. The samples were sent to our official laboratory and registered following all procedures according to legal provisions (custody and privacy).

### 2.3. Sample Extraction

Before extraction, samples are milled to reduce the particle size and improve the accessibility of residues enclosed in the interior of the materials (e.g., <500 µm) (Grindomix GM200, Retsch, Pedrengo, Italy).

The extraction procedure is based on the QuPPe method [20] with some modifications. Briefly, 5.00 ± 0.05 g were weighed into a 50 mL centrifuge tube and spiked with an internal standard (perchlorate ^18^O_4_). The sample was rehydrated with 10 mL H_2_O. Then 5 mL MeOH + 1% HCOOH was added. It was shaken using a standardize device Agytax (Lab Service Analytica srl, Anzola dell’Emilia, Italy) setting the following parameters: breadth 180 mm; velocity 2.0 msec^−1^; acceleration 45 m s^−2^; jerk 7 level, delay 0.05 s and duration 900 s. Thereafter, the extract was centrifuged at 4500 rpm for 10 min at 4 °C (Megafuge 16 Centrifuge Series, Thermo Fisher). The supernatant was transferred in a 15 mL centrifuge tube and frozen overnight.

The extract was centrifuged again at 4500 rpm for 10 min at 4 °C and filtered with a 0.22 µm PVDF syringe filter (Merck Millipore Ltd., Darmstadt, Germany) and diluted 1:1 (*v*/*v*) in H_2_O before the IC-HRMS analysis.

### 2.4. IC-HRMS Analysis

The chromatographic system consisted of an ultra-high-performance ion chromatography (U-HPIC) instrument, Dionex™ ICS-5000 HPIC™. Figure 1 reports the configuration of the IC-HRMS system. Chromatographic separations were carried out using an analytical column Thermo Scientific™ Dionex™ IonPac™ AS19 (2 × 250 mm, with particle diameter of 4 µm) with a pre-column Thermo Scientific™ Dionex™ IonPac™ AG19-4µm RFIC™ (2 × 50 mm).

The oven and autosampler temperature were set, respectively, at 30 and 18 °C. The injection volume was 10 μL. The mobile phases consisted of 100 mM sodium hydroxide (NaOH) (phase A) and Water (phase B). The analysis was done at a flow rate of 0.250 mL min^−1^ using the following gradient elution: at the beginning, 30% phase A was constant for 5 min, and it was increased to 50% in 2 min. The latter was maintained for 8 min and increased to 100% in 2 min. This step was maintained for 7 min, and then switched back to the initial 30% in 0.5 min and kept constant for 10.5 min giving a total runtime of 35 min. The NaOH eluent was neutralized using a Dionex ADRS 600 e 2 mm dynamically regenerated suppressor (Thermo Scientific, Sunnyvale, CA, US).

The NaOH eluent is not compatible with the mass spectrometer. Thus, we used a postcolumn eluent suppressor device that electrolytically converts the hydroxide to water and removes the sodium counterions from the system. We used an organic modifier, MeCN + 0.02% HCOOH in our case, to assist the desolvation of water in the mass spectrometer. This process requires an auxiliary pump, but the benefits make it worth doing.

The U-HPIC system was connected to a single-stage Orbitrap mass spectrometer, Q Exactive™, from ThermoFisher Scientific (Bremen, Germany), through a heated electrospray interface (HESI-II) operating in negative ionization [41].

The HESI parameters in negative polarity were the following: electrospray voltage of 3.8 kV; sheath gas of 35 arbitrary units; and auxiliary gas of 10 arbitrary units; capillary temperature 250 °C and auxiliary temperature 230 °C. The analysis was performed in Parallel Reaction Monitoring (PRM) mode. The PRM runs were acquired with a resolving power of 70,000 FWHM for the parental ions, AGC target of 2 × 10^5^, and max IT 100 ms. All the chromatographic runs were carried out using a stepped energy collision of 30, 40, and 60 eV for all compounds included in the method. Instead, the bromide ion has an energy collision of 10 eV; for perchlorate and chlorate, 200 eV.

### 2.5. Method Validation

The SANTE/11312/2021 guidance document on analytical quality control and method validation procedures for pesticides residues in food and feed was used to verify that the method was fit for purpose [42].

The validation of the present method was carried on a rice matrix for the commodity group of cereals and legumes. The rice matrix was spiked at four concentration levels for five replicates. The selected spiked levels were [0.005 mg kg^−1^], [0.010 mg kg^−1^], [0.10 mg kg^−1^], and [1.0 mg kg^−1^]. The assessed parameters were linearity, lowest calibration level (LCL), limit of quantification (LOQ), repeatability, reproducibility (RSD_W_R__), exactness, and recovery.

The method was assessed by means of a proficiency test (PT), EUPT-SRM16 (milled sesame seeds). The uncertainly parameter, in accordance with the SANTE/11312/2021 document, was evaluated using quality-control tests (spiked samples) for the cereal commodity group.

### 2.6. Data Analysis

IC-HRMS data were elaborated using TraceFinder 3.3 software. The bromide ion was internally standardized against perchlorate ^18^O_4_.

## 3. Results and Discussion

### 3.1. Method Development

The objective of this study was to evaluate the possibility of using an ion chromatography–high-resolution mass spectrometry (IC-HRMS) application for fast routine analysis of bromide ion in cereals and legumes.

Traditionally, the bromide ion was determined as total inorganic bromide by GC-ECD, which involved a prior derivatization analysis. Afterwards, new methods were developed using liquid and ionic chromatography-coupled low-resolution mass spectrometry [20,43]. Bromide is composed of two naturally stable isotopes, 79Br¯ and 81Br¯, present with a relative mass abundance of 50.686%. No MS/MS fragmentation is possible. So, the analysis has to rely on “parent/parent” determination. As reported by Eichhorn et al. (2016) [43], during the 11th European Pesticide Residue Workshop in Cyprus, the determination of bromide is affected by interferences as phosphonic acid (*m/z* = 81). This analyte, which is naturally present in many samples, could affect the bromide ion determination, depending on the chromatography column’s efficiency.

The strategy to overcome the interferences of MS/MS analysis, monitoring *m/z* = 79/79 and *m/z* = 81/81, involves the dilution of the sample to reduce the matrix effect and the improvement of the chromatographic separation. Another way to reduce the interferences of the phosphonic acid is to increase the collision energy, as reported by Eichhorn et al. (2016) [43].

The proposed method in this work allows the determination bromide ion using high-resolution mass spectrometry. The bromide ion was internally standardized against perchlorate ^18^O_4_. This labelled internal standard was selected because this is a multi-analysis method that include the determination of phosphonic acid, fosetyl-Al, chlorate, and perchlorate.

The application of IC-HRMS can overcome the issues experienced with other chromatographic techniques for the analysis of QuPPe extracts. The mass trace *m/z* 80.9154/80.9154 is highly recommended for quantifications whereas *m/z* 78.9174/78.9174 can be used as a qualifier ion. The mass trace *m/z* 80.9154 was interfered with by phosphonic acid, at [H_2_PO_3_]^−^ = 80.9780, whereas *m/z* = 79 with 78.9576 *m/z* (Figure 2a). High-resolution mass spectrometry could overcome the interference problem; the Δppm of the mass 80 *m/z* is 773 and for the 79 *m/z* is 509. It is not possible to obtain the same results with low-resolution mass spectrometry. The retention time could also help to identify the two different analytes: the bromide ion at 5.87 min and phosphonic acid at 5.18 min (Figure 2b). To be taken into account, is that the extraction method, which uses acidified methanol, does not involve a clean-up step, and the extracts contain high concentrations of matrix co-extractives. In particular, for legumes that contain a high percentage of proteins, the frozen step is crucial to avoid problems with the plugging of the chromatography column and consequently the shift in retention time.

The samples were also diluted with water (1:1, *v*/*v*) to help the identification, minimize the matrix effect, and to avoid bad peak shapes. Lopez et al. (2019) also reported the importance of a large dilution of the sample extract before injection to avoid the degradation of the column and to reduce the matrix effect and peak splitting [19].

### 3.2. Method Validation

In order to verify the suitability of the method for the scope of routine application, the main points of the SANTE/11312/2021 document were followed [42]. The method validation was carried out only on the rice matrix, considering the difficulty to find a blank one. For this reason, it was not possible to do the evaluation of the specificity using blank reagent and blank control samples, as reported in the SANTE document, in the validation parameters and criteria section. However, the method demonstrates the capability to identify the bromide ion and discriminate its interference (phosphonic acid) using the retention time (RT) and the Δppm thanks to the application of the high-resolution mass spectrometry. The calibration criteria were set at an R^2^ value ≥0.98 in the considered concentration range, recoveries in a range of 70–120%, and residuals for the calibration graph within ±20%.

The rice matrix was weighted and spiked at four concentration levels for five replicates. The spiked levels were [0.005 mg kg^−1^], [0.010 mg kg^−1^], [0.10 mg kg^−1^], and [1.0 mg kg^−1^].

The MRL values reported for bromide ion in different food commodities were amply scattered, from 20 to 50 mg kg^−1^, for the selected commodity group. It was decided to test the low concentration to verify the sensitivity of the method [44]. However, for a cereal commodity, the recoveries at the spiked concentration below [0.10 mg kg^−1^] did not pass the validation because the blank matrix naturally contains bromide residues. The spiked samples at [0.005 mg kg^−1^] and [0.010 mg kg^−1^] did not significantly increase the signal and it was not possible to correctly quantify. Instead for the concentration higher than [0.10 mg kg^−1^], recoveries were satisfactory, and the mean values ranged between 99 and 106%, with a relative standard deviation lower than 3%. The linearity was evaluated in the matrix in a range between 0.010 and 2.5 mg kg^−1^, with a coefficient of determination R^2^ = 0.9962. The lowest calibration level (LCL) in the blank matrix was 0.01 mg kg^−1^. The exactness, determined using the % bias, was calculated with the percentage average recovery of the spiked samples. At [0.10 mg kg^−1^] the % bias was 6; at [1.0 mg kg^−1^] it was −1.

The LOQ for the selected commodity is [0.10 mg kg^−1^]. It is not necessary to go down with LOQ, considering that bromide, as mentioned previously, is naturally present in food due to its presence in the soil. The matrix effect was not evaluated, and the calibration curve was built in the matrix considering that the shift in retention time is influenced by the co-extractives. In specific cases of suspected violative residues or for difficult matrices, the final quantitative determination was carried out using the multiple standard addition approach. The European Union Reference Laboratories (EURLs) have to organize Proficiency Tests (EUPTs). According to legislation 396/2005/EC [38], all laboratories analyzing samples within the framework of official controls on pesticide residues shall participate in the community proficiency tests for pesticide residues organized by the DG-SANCO.

The performance of the proposed method was evaluated by participating in EU-PT for single-residue methods, in 2021 (EUPT-SRM16- milled sesame seeds). In this PT, the bromide ion concentration determined by our laboratory was 19.0 mg kg^−1^ against the assigned value of 21.3 mg kg^−1^, with an acceptable z-score of −0.4. Furthermore, the phosphonic acid bromide ion interference was present in the test item and it was also reported at a concentration of 0.625 mg kg^−1^ against the assigned value of 0.676 mg kg^−1^. It was correctly quantified, with a z-score of about −0.3.

The uncertainly of the method was evaluated using the quality control tests (spiked samples) for the cereal commodity group and, in accordance with the SANTE 11312/2021 document, it was demonstrated to not exceed the 50% default value (measurement of the expanded reproducibility uncertainty, U’ = 32%, with the expanded coverage factor k = 2).

### 3.3. Sample Analyses

The proposed method was applied to 34 cereals and legumes and the results are reported in Table 1.

It is important to bear in mind that the soil may contain bromide ion naturally or from anthropogenic source (fertilizers and pesticides that contain bromide or previous fumigations) and it is not easy to establish its provenience. The average value of bromide ion in cereals is about 5.5 mg kg^−1^ and in legumes 6.3 mg kg^−1^, excluding Chickpeas_Flour_b (non-compliant). Instead, the median value for cereals of 2.6 mg kg^−1^ versus legumes is about 4.4 mg kg^−1^. The legumes, such as lentils, chickpeas, and beans, presented a composition with a high percentage of proteins (approximately 20%) that promotes the absorption of bromide ions.

Among the legume samples, the chickpea flour (Chickpeas_Flour_b) was collected and sent to our laboratory from border inspection posts in a shipment of about 864 kg. This sample was non-compliant given the concentration of bromide ion (117 mg kg^−1^) probably due to the fumigation treatment carried out to preserve the food integrity during the trip. The Regulation (EC) 839/2008 reported an MRL of 30 mg kg^−1^ for the chickpea matrix [44]. In this specific case of non-compliance, the determination was carried out also using the multiple standard addition approach. Five portions of the sample (Chickpeas_Flour_b) were weighed and three of them were spiked: at the concentration of the first screening, half, and 1.5-fold. Each sample was injected two times. The graphical presentation is shown in Figure 3 via linear regression. The average concentration calculated using this approach was 117.204 ± 58.602, in accordance with the first screening using the matrix calibration curve. In accordance with the SANTE/11312/2021 document [42], the relative difference of the two replicates of the individual results should not exceed 30% of the mean.

A first overview of the bromide background levels collected by EFSA over the last years showed a lack of data about its concentration in different commodities. For this reason, the European Commission has included in its national monitoring program the bromide ion in several commodities in order to collect data, especially on those commodities for which data are still lacking, so that the database can be completed [39]. Regulation (EU) 2021/601 [40] requires the determination of this ion in lettuces and tomatoes in 2022, and for brown rice matrices in 2023.

## 4. Conclusions

Current international crises, the COVID-19 restrictions, and climate change have impacted food security and created a new scenario for global food production and supply. A new method to determine bromide ion residues in cereals and legumes, using suppressed ion chromatography coupled to high-resolution mass spectrometry (IC-HRMS), was developed. The method was validated in accordance with the SANTE 11312/2021 document to verify its suitability in the scope of routine application. It was applied to different samples (n = 34) and tested in the EUPT-SRM16 with satisfactory results.

By routinely applying the proposed method, data will be collected to increase the knowledge about the background levels of bromide ions in the selected matrices. This is only a starting point to build an overview of the bromide ion scenario. All data that will be collected could help the legislator to estimate the range of bromide ion concentrations naturally present in food and define and re-evaluate the MRL and toxicological value. The proposed method could also help to monitor compliance with respect to the regulations in force for foods in transit.

## Figures and Tables

**Figure 1 foods-11-02385-f001:**
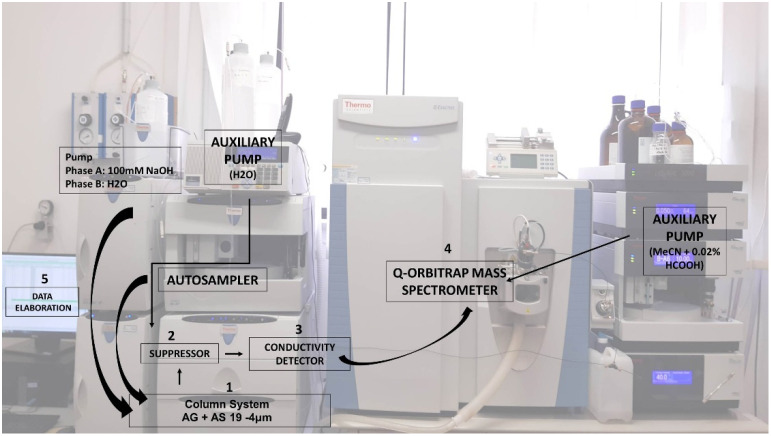
Configuration of the IC-HRMS system: (1) the column system (precolumn and chromatographic column); (2) suppressor; (3) conductivity detector; (4) mass spectrometer (Q-Orbitrap); and (5) data elaboration.

**Figure 2 foods-11-02385-f002:**
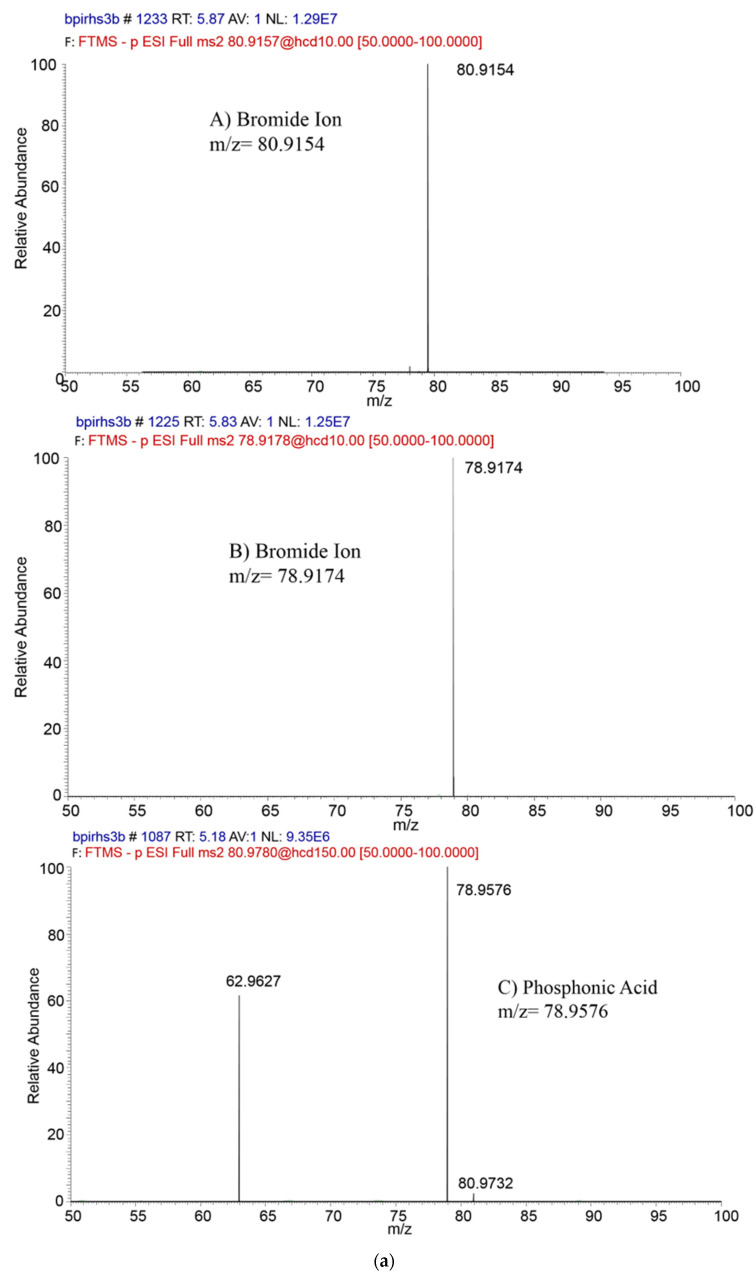
Identification of bromide ion and phosphonic acid: (**a**) PRM acquisition in ESI-: bromide ion quantifier, *m/z* = 80.9154; bromide ion qualifier, *m/z* = 78.9174; and phosphonic acid, *m/z* = 78.9576; (**b**) chromatogram of the bromide ion at RT = 5.87 min (quantifier and qualifier) and phosphonic acid at RT = 5.18 min.

**Figure 3 foods-11-02385-f003:**
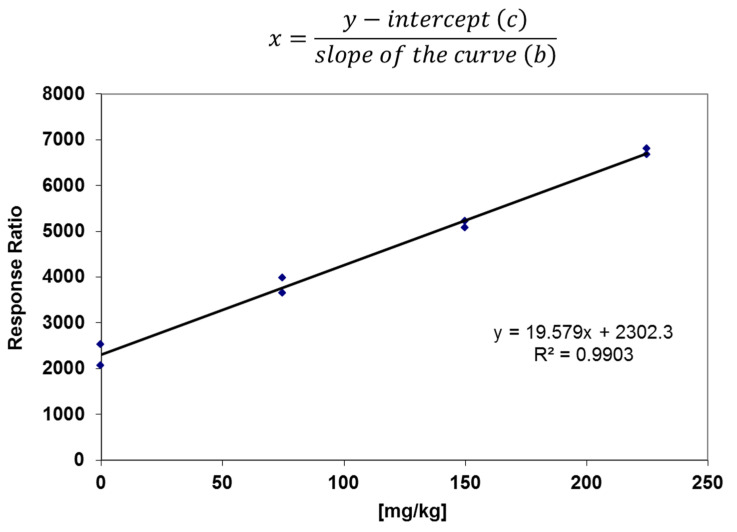
Graphical presentation of the linear regression of the Chickpeas_Flour_b sample.

**Table 1 foods-11-02385-t001:** List of samples analyzed for bromide ions.

	Matrix	Production	Bromide Ion (mg kg^−1^)
LEGUMES	Beans_A	EU	3.6
	Beans_B	EU	<0.10
	Beans_C	EU	<0.10
	Beans_D	EU	<0.10
	Beans_Black	EU	1.4
	Chickpeas_A	EU	20
	Chickpeas_B	EU	2.8
	Chickpeas_Flour_a	EU	5.1
	Chickpeas_Flour_b	India	117
	Lentils_A	EU	6.4
	Lentils_B	EU	1.3
	Lentils_Red	EU	13
	Lentils_Black	EU	3.2
CEREALS	Barley pearl	EU	3.5
	Corn	EU	1.1
	Oats	EU	1.8
	Quinoa	EU	2.4
	Rice_A	EU	14
	Rice_B	India	20
	Rice_C	India	1.8
	Rice_D	India	15
	Rice_E	EU	<0.10
	Rice_F	EU	13
	Rice_G	EU	14
	Rice_Brown	EU	<0.10
	Rice_Carnaroli	EU	0.12
	Rice_Vialone Nano	EU	0.10
	Rice_Basmati	EU	0.41
	Rice_Venere	EU	<0.10
	Wheat_A	EU	2.8
	Wheat_B	EU	6.6
	Wheat_C	EU	2.1
	Mix of cereals	EU	<0.10

## Data Availability

Data is contained within the article.

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
