# Peer review of "Ion Chromatography–High-Resolution Mass Spectrometry Method for the Determination of Bromide Ions in Cereals and Legumes: New Scenario for Global Food Security"

_foods, 2022, doi:10.3390/foods11162385_

Round 1
Reviewer 1 Report
The authors of the article raise an important problem of the content of bromide ions in food of plant origin (cereals, rice) resulting from the presence of these compounds in the soil on which they grew.
The publication presents e new method to determine bromide ion residues in cereals and legumes, using suppressed ion chromatography coupled to high resolution mass spectrometry (IC-HRMS), wich was developed by authors.
The article is written in plain language. It contains all the descriptions of experiences necessary for understanding. The presented results are well and critically described by the authors. The work is also supported by the current literature on the presence and determination of bromide ions in food.
After reading the work, the following questions and comments come to my mind:
1. Chromatograms and mass spectra in the drawings at work are unreadable
2. If the authors observed the effect of the matrix, why did they not use the multiple standard addition technique when determining the content of analytes? The use of this technique at the extraction level would take into account the influence of the matrix on the extraction process itself as well as the final determination of the content of compounds in the tested samples
3. What is the limit of detection?
4. How confident are the authors that the LOQ will be the same for all matrices?
In my opinion, the work has potential for the future.
Author Response
Q1. Chromatograms and mass spectra in the drawings at work are unreadable
A1. Done
Q2. If the authors observed the effect of the matrix, why did they not use the multiple standard addition technique when determining the content of analytes? The use of this technique at the extraction level would take into account the influence of the matrix on the extraction process itself as well as the final determination of the content of compounds in the tested samples
A3. We agree with your observation. The multiple standard addition technique requires a first semi-quantitative screening to determine the presence and the concentration level of the compound and then the final quantitative determination.
The aim of our work is to investigate a huge number of samples in order to collect data about the background level of bromide ion. Considering the uncertainly of the method (U’=32%) and the satisfactory results of the EUPT-SRM16 using a blank matrix calibration curve, the method with the multiple standard addition technique is applied only in the specific cases of suspected violative residue or for difficult matrices.
Q3. What is the limit of detection?
A3. Considering the difficulty to find a blank matrix, the low detection (LOD) is not relevant and for this reason it was not reported in the paper. Instead the lowest calibration level (LCL) in matrix was reported.
Q4. How confident are the authors that the LOQ will be the same for all matrices?
A4. No, we are not sure, but it is not possible to determine the LOQ in all different matrices considering the difficulty to find for each of them a blank to spike.

Reviewer 2 Report
In the manuscript, ion chromatography high resolution mass spectrometry method was developed for the determination of bromide ion in cereals and legumes. The topic is important to protect food safety. However, there are many publications similar to the topic, is there difference or novelty with other publications? And, there is no more detailed discussion with other detection methods. While not really bad, the English needs to be improved at whole article.
Specific comments:
In first paragraph of introduction, why propose the COVID-19? is there relationship with residues of bromide ion?
Author need to provide higher resolution figures.
How to obtain the real samples with ions? please give more description in method part.
Author Response
Q1) In the manuscript, ion chromatography high resolution mass spectrometry method was developed for the determination of bromide ion in cereals and legumes. The topic is important to protect food safety. However, there are many publications similar to the topic, is there difference or novelty with other publications? And, there is no more detailed discussion with other detection methods. While not really bad, the English needs to be improved at whole article.
A1) Traditionally, in food field, the bromide ion was determined as total inorganic bromide by GC-ECD that involved the derivatization prior analysis. Afterwards, new methods were developed using liquid and ionic chromatography coupled low resolution mass spectrometry as reported in QuPPe Document. The novelty of our method is the determination of the bromide by high resolution mass spectrometry to overcome the problem of interferences, such as phosphonic acid. In the paper, the word “fast” means that the analysis without the presence of interferences is a simplifier determination.
The aim of our work was to propose an analytical method to apply in the routine analysis in order to build an overview of the bromide ion scenario.
As suggested, the paper was implemented including in the introduction the background of the determination of bromide using different detection methods in foods. The discussion of the method development was also modified.
Specific comments:
Q2) In first paragraph of introduction, why propose the COVID-19? is there relationship with residues of bromide ion?
A2) There is not a direct relationship between the viral infection and the residues of bromide ion, but the market trade was affected by the global situation. The text was modified to provide clarification.
Q3) Author need to provide higher resolution figures.
A3) Done
Q4) How to obtain the real samples with ions? please give more description in method part.
A4) The method part about the sample information was modified. All samples were collected in the framework of official control activity and from border inspection posts. The samples are sent to our official laboratory and registered following all procedures according to legal provisions (custody and privacy).

Round 2
Reviewer 2 Report
Agree to accept